A retrospective study detects a novel variant of porcine epidemic diarrhea virus in England in archived material from the year 2000

Steinbach Falko falko.steinbach@apha.gsi.gov.uk 1
Dastjerdi Akbar 1
Peake Julie 1
La Rocca S. Anna 1
Tobin Frank P. 2
Frossard Jean-Pierre 1
Williamson Susanna 3
1 Department of Virology, Animal and Plant Health Agency , Addlestone , Surrey , United Kingdom
2 Holmefield Farm Services Ltd , Murton , Yorkshire , United Kingdom
3 Surveillance Intelligence Unit, Animal and Plant Health Agency , Bury St Edmunds , Suffolk , United Kingdom
Jiao Peirong
Electronic publication date: 2016 Oct 27
Publication date: 2016
Volume: 4
Electronic Location ID: e2564
Received 2016 Jul 11; Accepted 2016 Sep 13
Copyright: ©2016 Steinbach et al.
Copyright year: 2016
Copyright holder: Steinbach et al.
License: This is an open access article distributed under the terms of the Creative Commons Attribution License, which permits unrestricted use, distribution, reproduction and adaptation in any medium and for any purpose provided that it is properly attributed. For attribution, the original author(s), title, publication source (PeerJ) and either DOI or URL of the article must be cited.
License URL: https://creativecommons.org/licenses/by/4.0/

Keywords: England, Pig, Diarrhea, PEDV, Variant, Phylogeny

Funding: Scanning surveillance for Pig Diseases in England and Wales ED1200 (Defra) Defra project “Scanning surveillance for Pig Diseases in England and Wales” (ED1200) funded laboratory and staff costs of this work. The funders had no role in study design, data collection and analysis, decision to publish, or preparation of the manuscript.

==============================
Outbreaks of porcine epidemic diarrhea (PED) were first recorded in England in the 1970s and continued to be confirmed until 2002. Retrospective analysis of archived material from one of the last confirmed cases in England in the year 2000 demonstrates the previous existence of a very diverse PED virus strain. Following the outbreaks of PED in North America in 2013, there has been renewed interest in phylogenetic analysis of sequences from PEDV strains worldwide. There is a gap in the available sequence data between the mid 1980s and the mid 2000s. This work is an example of how this gap can be at least partially filled by the examination of archived material.

Introduction

Porcine epidemic diarrhea virus (PEDV), an Alphacoronavirus in the order Nidovirales, was first described in England in the 1970s and subsequently emerged across most of Europe (Wood, 1977; Pensaert & De Bouck, 1978). While the virus reached southeast Asia in the 1980s, reports of PED declined in Europe and no cases have been reported from the field in England since 2002 (Williamson et al., 2013). In recent years, however, variant strains of PEDV have emerged in China causing high mortality in sucking piglets and spreading rapidly between farms (Sun et al., 2012). In 2013 these virulent PEDV strains were reported in pig herds in the United States (Stevenson et al., 2013). This was followed by a report from Ohio of another PEDV strain characterised by insertions/deletions in the S-gene, known as the INDEL strain, OH851 being the prototype (Wang, Byrum & Zhang, 2014), associated with reportedly milder disease in the field. Most recently INDEL PEDV strains have been detected in outbreaks of diarrhoea in Western Europe, including Germany, the Netherlands, France and Italy (Hanke et al., 2015; EFSA, 2016). In parallel, the PEDV strain associated with higher virulence in Asia and North America was detected in the Ukraine (Dastjerdi et al., 2015).

The molecular characterization of PEDV has been established (Huang et al., 2013). The virus is an enveloped, positive-sense, single-stranded RNA coronavirus possessing a genome of ≈28 kilobases. To classify strains involved in the previous and recent outbreaks, a phylogenetic nomenclature using genogroups or clades has been suggested (Huang et al., 2013). Markers for pathogenicity are poorly defined for coronaviruses, thus any focused sequencing of the S1 region only, for example, might introduce unreasonable bias into the analysis.

This paper reports the results of analysis of the PEDV from a historic field case in England, including comparisons with currently circulating North American/Asian viruses, INDEL strains and the European prototype strain CV777 (Kocherhans et al., 2001).

Materials and Methods

Clinical case history and diagnostic investigation

Between 1999 and 2002, the national veterinary diagnostic database for GB (VIDA) recorded 13 diagnoses of PED on nine pig units in five counties of England, with the last diagnosis being made in 2002 and none since. Unfortunately, the archived data does not allow us to draw epidemiological links between these cases. Archived material was only available from one outbreak from October 2000; a finishing unit in Yorkshire reporting an outbreak of diarrhoea. Three fecal samples were submitted at that time, and a diagnosis of PED was made by testing the samples using a duplex RT-PCR (primer sequences TTTATTCTGTCACGCCATGT and CCAGATTTACAAACACCTATGTTA spanning a 199 bp fragment) designed to detect and discriminate the S gene from both PEDV and transmissible gastroenteritis virus (TGEV). All three samples tested positive for PEDV. Differential diagnostic testing for Salmonella and Brachyspira species by selective culture identified Salmonella enterica ser. Typhimurium phage type 193 only after enrichment culture in one of the three fecal samples, negating a diagnosis of salmonellosis or swine dysentery. One of the samples, from a 14-week-old pig, was retained in the cryostore archive, and was retrieved for further analysis in 2014.

Analysis of the archived PEDV-positive faeces

The original positive PEDV PCR result was re-confirmed on the archived faeces, and also using a further in-house PCR targeting the N gene. The initial PCR amplicon was subjected to conventional Sanger sequencing to confirm the detection of PEDV. Upon recognition of the significant differences to other PEDV strains, next generation full genome sequencing was carried out to avoid the necessity of designing novel primers in multiple rounds of Sanger sequencing for such a large RNA genome. In brief, the extracted PEDV RNA was subjected to DNase digestion and used as template for cDNA generation using a cDNA Synthesis System (Roche) for preparation of sequencing libraries using a Nextera XT kit (Illumina, San Diego, CA, USA). Paired end sequencing was performed on an Illumina MiSeq. Finally, the consensus sequence was obtained by de novo assembly using Velvet 1.2.10 as previously described (Miller, Koren & Sutton, 2010) and re-evaluated using the templated genome assembly function of the SeqMan NGen v13 software (DNASTAR Inc. Madison, WI, USA). The consensus sequence was generated from 321425 sequence reads; considering length of each read the average coverage for each base pair is 1341.15, which is well above the optimum coverage. For phylogenetic analysis, comparisons were made with published spike protein amino acid sequences of 32 other PEDV strains and two other coronaviruses, and similarly for the ORF1a/b nucleotide sequence. Sequence alignments were performed using the Clustal W algorithm, and phylogeny was performed using the maximum likelihood method with 1,000 bootstrap replications, both using MEGA version 6 (Tamura et al., 2013).

Results

PCR testing of the faecal sample from 2000 confirmed the presence of PEDV nucleic acid and the absence of TGEV, indicating involvement of PEDV in the outbreak of diarrhoea in the finishing pigs.

As shown in Fig. 1, phylogenetic analysis of the consensus whole genome sequence obtained directly from the archived faecal sample indicates that the virus, while clearly belonging to the PEDV species, lies distinct from any of the genogroups described by Huang et al. (2013).

Figure 1 Molecular phylogeneticanalysis of PEDV sequences.

(A) Molecular phylogenetic analysis of 33 PEDV spike protein amino acid sequences. The evolutionary history was inferred by using the Maximum Likelihood method. The tree with the highest log likelihood (−10946.4157) is shown. Initial trees for the heuristic search were obtained by applying the Neighbor-Joining method to a matrix of pairwise distances estimated using a JTT model. A discrete Gamma distribution was used to model evolutionary rate differences among sites (5 categories (+G, parameter = 1.4131)). The rate variation model allowed for some sites to be evolutionarily invariable ([+I], 0% sites). (B) Molecular phylogenetic analysis of 33 PEDV ORF1 nucleotide sequences. The evolutionary history was inferred by using the Maximum Likelihood method based on the Tamura-Nei model. The tree with the highest log likelihood (−85920.7638) is shown. Initial tree(s) for the heuristic search were obtained by applying the Neighbor-Joining method to a matrix of pairwise distances estimated using the Maximum Composite Likelihood (MCL) approach. The tree is drawn to scale, with branch lengths measured in the number of substitutions per site. The trees are drawn to scale, with branch lengths measured in the number of substitutions per site. The percentage of trees in which the associated taxa clustered together is shown next to the branches. Evolutionary analyses were conducted in MEGA6 (Tamura et al., 2013). The GenBank accession numbers, country and year of isolation are indicated, along with the genogroups as described by Huang et al. (2013). Porcine respiratory coronavirus (PRCV) and transmissible gastroenteritis (TGEV) virus sequences are shown for reference. The prototype PEDV strains CV777, OH851, Colorado 2013 and England-1-2000 strains are underlined.

At the amino acid level, as shown in Table 1, proteins from this England-1-2000 strain vary in similarity from 91.5% to 96.3% compared to those of the CV777 strain. At the nucleotide level, similarities for the ORF1a/b region vary from 95.1% to 96.3% when compared to the other PEDV sequences analysed. While this strain is significantly different to other PEDV strains known so far, it is in no part more closely related to other known coronaviruses, thus does not resemble the result of inter-species recombination such as described for the Italian swine enteric coronavirus (Boniotti et al., 2016). The complete genome sequence of PEDV England-1-2000 has been deposited at GenBank under the accession number KU836638.

Discussion

While PEDV England-1-2000 was significantly different from the European prototype CV777 from the early 1980s (Ducatelle et al., 1981), current and established PCR tests in 2000 were able to detect the virus without modification, suggesting that similar strains were not simply missed due to lack of diagnostic potential. The virus also differs from the virulent North American/Asian and the INDEL variants such as OH851 currently circulating in Western Europe. The lack of other published PEDV sequences from historic outbreaks of PED to the 2000s prevents further comparisons, but this clearly represents a genotype not described previously.

Table 1 Amino acid identity of PEDV England-1-2000 compared to reference strains of European and US PEDV.

Protein	ORF1a/b	S	ORF3	E	M	N	
% identity to CV777 (AF353511)	96.3%	92.0%	91.5%	93.4%	94.2%	94.8%	
% identity to Colorado 2013 (KF272920)	96.2%	89.7%	91.1%	92.1%	93.4%	93.4%	
% identity to OH851 (KJ399978)	96.2%	91.7%	91.5%	92.1%	93.4%	93.2%	

No detailed morbidity or mortality data are available from records of the PED outbreak from which this virus was identified, but a previous diagnosis of PED was made on the farm in an earlier batch of finishing pigs in November 1999. The isolation of Salmonella Typhimurium in just one of the three fecal samples by enrichment culture only, and lack of isolation of Brachyspira species from the samples, indicates no significant involvement of salmonellosis or Brachyspira-related colitis in this diarrhea outbreak. In general, PED outbreaks at that time caused watery diarrhea without mortality in growing pigs and sows rather than piglets, and tended to spread rapidly through the herd. This clinical presentation may, in part, reflect immunity through natural infection in the national herd at that time. Unfortunately, no viable virus could be recovered from the limited amount of material available after the prolonged storage period to assess the relative virulence of this strain in experimental infections.

This finding shows the presence of a diverse field PEDV strain in England at the turn of the century, the pathogenicity of which requires further investigation that will only be possible upon rescuing the strain via reverse genetics. Outbreaks of PED were diagnosed in GB until 2002 although the annual incidence of laboratory confirmed outbreaks was quite low, and there is no evidence that this strain is still present as a cause of disease in GB (EFSA, 2016). This report highlights the merit of further investigation of archived material from Europe and elsewhere to establish the degree of heterogeneity of historic PEDV strains and determine whether this England-1-2000 strain was typical of those circulating at the time in Europe.

We are grateful to Sonia Zuñiga and Luis Enjuanes, Madrid for assisting attempts to isolate this strain, and to Richard J. Ellis for NGS sequencing.

Additional Information and Declarations

Competing Interests

Author Contributions

Data Availability

The authors declare there are no competing interests.

Frank Tobin is an employee of Holmefield Farm Services Ltd, Murton, Yorkshire, United Kingdom.

Falko Steinbach and Susanna Williamson conceived and designed the experiments, wrote the paper, reviewed drafts of the paper.

Akbar Dastjerdi conceived and designed the experiments, analyzed the data, contributed reagents/materials/analysis tools, wrote the paper, reviewed drafts of the paper.

Julie Peake and S. Anna La Rocca performed the experiments, reviewed drafts of the paper.

Frank P. Tobin contributed reagents/materials/analysis tools, reviewed drafts of the paper.

Jean-Pierre Frossard analyzed the data, wrote the paper, prepared figures and/or tables, reviewed drafts of the paper.

The following information was supplied regarding data availability:

GenBank accession number KU836638.

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
