# Peer review of "A retrospective study detects a novel variant of porcine epidemic diarrhea virus in England in archived material from the year 2000"

_PeerJ, doi:10.7717/peerj.2564_

## Round 0.1 · original submission · Major Revisions

· Academic Editor

Major Revisions

Dear Jean-Pierre, please revise your manuscript according to the reviewers' advice.

·

Basic reporting

Porcine epidemic diarrhea (PED), caused by Porcine epidemic diarrhea virus (PEDV) ,is an emergently and economically important pig disease that attracts attention of the scientists worldwide. The PED history in each country is different. The authors detected a variant PEDV in three samples from finishing fecal samples by RT-PCR followed by sequencing and phylogentic analysis. Their findings supported that PEDV existed two years earlier than the report of outbreak in 2002, and that the England-2000 strain was diverse from CV777 and other PEDV strains in some EU countries. This conclusion was of scientific importance. The accession number of the genome sequence indicated it met requirements of Data Sharing Policy.The manuscript was well written and the figure was strongly related with the experiment and the conclusion.

Experimental design

The authors used the archieved samples for detection of PEDV by RT-PCR, the whole genome sequence was analyzed and compared with other PEDVs. The methodology sounds reasonable. The retrospective study is rational.

Validity of the findings

The major findings in this experiment were the presence of a PEDV, England-2000, in the samples collected in 2000 and were different from other strains. These conclusions were based on their PCR amplifications and sequence analysis and supported their primary hypothesis. The results can be reproducible if the samples are privided to other persons.

Comments for the author

The detection of a variant PEDV from the samples collected in the year of 2000 by authors updates the PED history in England and it is of scientifc significance. However, in the point of a acceptable manuscript,I would like to suggest the authors to further provide the following information in their revised manuscript. (1) Are there any epidemiological links between 13 PED cases in nine units if the authors happened to have these data (Line 51 to Line 52); (2) the primer pairs sequence in duplex RT-PCR and whether the fragement,S1, or the full S gene was amplified (Line 56). (3) Explain whether the "immunity" (Line 113) were acquired from vaccination or from any other measurements. If vaccination, then what kind of vaccine strain was used.

·

Basic reporting

Appears to meet standards

Experimental design

Appears to meet standards although a more detailed description of the robustness of the single sequence obtained by NGS and a more thorough analysis of the sequence obtained would strengthen the work.

Validity of the findings

Appears to meet standards although a more detailed description of the robustness of the single sequence obtained by NGS and a more thorough analysis of the sequence obtained would strengthen the work.

Comments for the author

A more detailed description of the robustness of the single sequence obtained by NGS and a more thorough analysis of the sequence obtained would strengthen the work. Suggest to do more phylogenies using nucleotide and amino acid alignments of more conserved regions, e.g. the ORF1a/b rather than only showing the phylogeny based on the Spike protein. Did they do any analysis to see whether recombination may play a role?
The manuscript contains inconsistent and often incorrect use of Capital letters when writing e.g. porcine epidemic diarrhea virus or coronavirus etc.
In the introduction the authors write that markers for pathogenicity are poorly defined and then go on to say that thus any molecular characterisation is best carried out by full genome analysis. Why that is the case is not clear as even the full sequence will not reveal unknown markers of pathogenicity and moreover, the authors do only include a very limited analysis of their full sequence and thus do not provide any compelling evidence for why their NGS approach is optimal or needed.
In the discussion the authors state that the presence of a diverse strain of PEDV in England around the turn of the century, "for which the potential impact on immunogenicity and control requires further study". Why this further study is required is not clear to this reviewer although it is clear that more analysis of historical/archived samples are clearly called for.

---

## Round 0.2 · accepted · Accept

· Academic Editor

Accept

Dear Jean-Pierre,thank you for your submission. Your revised manusript has been accepted.